# Compositional Effects of Additively Manufactured Refractory High-Entropy Alloys under High-Energy Helium Irradiation

**DOI:** 10.3390/nano12122014

**Published:** 2022-06-10

**Authors:** Eric Lang, Kory Burns, Yongqiang Wang, Paul G. Kotula, Andrew B. Kustas, Sal Rodriguez, Assel Aitkaliyeva, Khalid Hattar

**Affiliations:** 1Sandia National Laboratories, Albuquerque, NM 87185, USA; koryburns@ufl.edu (K.B.); pgkotul@sandia.gov (P.G.K.); akustas@sandia.gov (A.B.K.); sbrodri@sandia.gov (S.R.); khattar@sandia.gov (K.H.); 2Department of Nuclear Engineering, University of New Mexico, Albuquerque, NM 87131, USA; 3Department of Materials Science and Engineering, University of Florida, Gainesville, FL 32611, USA; aitkaliyeva@mse.ufl.edu; 4Los Alamos National Laboratory, Los Alamos, NM 87545, USA; yqwang@lanl.gov

**Keywords:** high-entropy alloys, refractory metals, helium bubbles, TEM

## Abstract

High-Entropy Alloys (HEAs) are proposed as materials for a variety of extreme environments, including both fission and fusion radiation applications. To withstand these harsh environments, materials processing must be tailored to their given application, now achieved through additive manufacturing processes. However, radiation application opportunities remain limited due to an incomplete understanding of the effects of irradiation on HEA performance. In this letter, we investigate the response of additively manufactured refractory high-entropy alloys (RHEAs) to helium (He) ion bombardment. Through analytical microscopy studies, we show the interplay between the alloy composition and the He bubble size and density to demonstrate how increasing the compositional complexity can limit the He bubble effects, but care must be taken in selecting the appropriate constituent elements.

## 1. Introduction

The harsh environment of future nuclear fission and fusion reactors will present new materials challenges via high temperatures, neutron, as well as low energy deuterium (D) and He particle bombardment [1]. For example, nuclear fusion reactor concepts require plasma-facing materials, which will be subjected to energetic helium ion bombardment in a range of energies from 100s of keV in the first wall to 10s of eV in the divertor [2]. Refractory metals, such as W, are proposed as plasma-facing materials for their high melting temperature, good thermal conductivity, and high sputter resistance. On the other hand, advanced nuclear reactor concepts, such as the Energy Multiplier Module being developed by General Atomics, involve fast neutrons. In this case, the neutrons have energies in excess of 0.1 MeV and an average neutron flux of 4.72 × 10^14^ n cm^−2^ s^−1^ [3], with the generation of highly energetic fission fragments, gamma rays, and alpha particles (He^2+^).

Helium implanted into metals is insoluble and precipitates out as bubbles in the matrix. At the high temperatures present in the reactor environment, the bubbles can migrate and coalesce into defects. All of these effects on the material microstructure detrimentally impact the bulk material properties and performance and shorten the component lifetime—a particular concern with the refractory metals proposed for use in these conditions [4]. The accumulation of He bubbles can embrittle tungsten and other refractory metals such as Nb, cause swelling and hardening [5,6], and at high temperatures can cause the growth of surface nanostructures, depending on the irradiation conditions [7,8,9]. The low thermal conductivity, change in electron emission, and potential for enhanced erosion of the helium-induced surface nanostructures raise concerns of dust formation, enhanced erosion, and unipolar arcing [10]. This has motivated multiple investigations into materials that limit or prevent this fuzz formation. Therefore, the impacts of He irradiation on the microstructure must be mitigated to maintain a predictable material performance in the harsh operational environment.

However, material microstructures can be engineered to tolerate the helium irradiation environment to mitigate the effects of He bubble accumulation. Strategies for limiting helium bubble effects in metals include decreasing the grain size [11,12], intentionally controlling the columnar microstructure [13,14], alloying [15,16], and introducing second phase interfaces/interphases [17,18,19]. Trapping He bubbles at the second phase precipitates or at interfaces in layered composites can mitigate the He effects in the matrix and may be more stable at high temperatures. Grain boundaries are strong traps for He bubbles and increasing the grain boundary density has been shown to reduce the number of intergranular He bubbles, thus reducing the swelling risk [12,20]. However, He accumulation will change the mechanical properties; additionally, nanocrystalline metals with a high grain boundary (GB) density are unstable at high temperatures and prone to grain growth [21,22].

To address the issues nanocrystalline materials face at high temperatures, the alloying of W has shown some promise in reducing the effects of He. Alloying W with V has computationally been shown to limit He diffusion in W [23]. W-Ti alloys have shown reduced fuzz growth when subject to He plasmas, attributed to the refined grain size [16]. W-Ta alloys have shown an increased threshold for fuzz formation [24]. W-Nb alloys have shown enhanced He bubble formation in Nb-rich regions [25], while W-5%V alloys mitigated fuzz formation compared to pure W [26]. W-based quinary alloys have also been fabricated, and their mechanical properties investigated [27].

A new class of materials proposed for use in future extreme environments are High-Entropy Alloys (HEAs), which are multicomponent (more than four elements, often in equiatomic concentrations) materials [28]. Refractory HEAs (RHEAs) may combine the benefits of refractory metals (high melting point, corrosion resistance) with the enhanced benefits of concentrated alloying to improve the mechanical and corrosion properties [29,30,31]. Conventional RHEAs are fabricated via conventional sintering powder metallurgy methods that include compaction followed by subsequent high temperature, which prolongs the fabrication time (due to the solid-state diffusion limits) and constricts the design space. Functional materials for future extreme environments require designs tailored for their application, which is not achievable through conventional sintering or hot-pressing methods. Additive Manufacturing (AM) is a rapid fabrication method that allows for unique component design, even in complex alloys such as RHEAs [29,32] One example of AM is laser beam directed energy deposition (LB-DED), enabling a rapid top–down near-net-shape fabrication of functionally tailored specimens [33,34]. AM techniques have even been proposed for fabricating compositionally and geometrically complex materials for use as plasma-facing materials in fusion reactors. However, the AM of refractory metals remains a challenge to create dense, uncracked parts without unmelted particles due to the high melting temperature, the mismatch in thermal expansion coefficients with build plates, and the intrinsic brittleness of the alloy that can be driven by high ductile-to-brittle transition temperatures. In order to overcome the issues with the AM of pure W, we pursue alloying to create dense microstructures with compositional complexity.

In this work, we explore the effect of refractory alloy composition on the helium bubble characteristics produced via AM. Two-to-four component refractory alloys are compared to pure W. We show that controlling the composition of refractory alloys, and RHEAs in particular, can improve their tolerance to He irradiation.

## 2. Materials and Methods

Refractory alloys were fabricated via LB-DED from elemental Mo, Nb, Ta, V, and W powders using an open-architecture LB-DED AM system [35]. Specimens were constructed from premixed elemental powders contained in separate powder reservoirs, targeting notionally equiatomic composition from NbTa, MoNbTa, MoNbTaW, and NbTaVW alloys, the latter two of which represent concentrated RHEAs with higher configurational entropy. For the NbTa and MoNbTa alloys, specimens were printed on a Ti substrate that was preheated at 200 °C so as to minimize specimen cracking and with a range of laser powers of 400–750 W and 220–350 W, respectively. These specimens were processed with a laser beam diameter of 1 mm and a laser scan velocity of 300–420 mm/min. The MoNbTaW RHEA was printed on a Mo substrate with a laser power of 1800 W, a beam diameter of 2 mm, and a laser scan velocity of 300 mm/min. Finally, the NbTaVW RHEA was processed on a Ti substrate that was preheated to 300 °C utilizing both a fusion and remelt laser pass at 800 W for the former and 1000 W for the latter, respectively. The laser velocity was 400 mm/min and 600 mm/min for the former and latter. A constant layer thickness of 250 µm was used. The pure W control sample is a commercially purchased sample. A bulk rolled 2 mm-thick W sheet was purchased, and 5 mm discs were cut from the sheet. All samples were polished to an EBSD quality finish with a colloidal silica vibratory polisher before He irradiation.

Samples were irradiated with 200 keV He^+^ ions at 900 °C to the fluence of 1 × 10^17^ cm^−2^ at a flux of ~3 × 10^13^ cm^−2^ s^−1^ on a 200 kV Danfysik Ion Implanter in the Ion Beam Materials Laboratory (IBML) at the Los Alamos National Laboratory (LANL). A thermocouple was mounted on a high-temperature alloy heater to measure temperature. The samples were mounted on the heater surface with silver paste to improve thermal transport between the heater and the samples. The target chamber vacuum was kept at ~1 × 10^−6^ Torr during the high-temperature He implantation. Stopping Range of Ions in Matter (SRIM-2013) software was used to estimate the ion damage and implantation profiles in the RHEAs [36]. Displacement energies of constituent elements were: Mo: 60 eV; Nb: 60 eV; Ta: 90 eV; W: 90 eV; V: 40 eV [37], and the density estimated via composition of the RHEA was determined via equiatomic estimations of processing.

Surface microstructure was investigated with a highly modified Scanning Electron Microscopy (SEM). JEOL JSM-IT300 SEM (Peabody, MA, USA) was operated at 30 kV utilizing an Elemental Dispersive Spectroscopy (EDS) via an EDAX Octane Elite Super detector. Transmission Electron Microscopy (TEM) studies were carried out with a modified JEOL JEM-2100 TEM operated at 200 kV, with bubbles imaged with ±500 nm Fresnel contrast. Scanning Transmission Electron Microscopy (STEM) studies were carried out with an FEI Titan G2–80-200 (Hillsboro, OR, USA) operated at 200 kV, equipped with a spherical aberration corrector, and the SuperX EDS detector with four silicon drift X-ray detectors with a combined solid angle of collection of 0.7 sr. TEM specimens were fabricated via FIB lift-out methods with an FEI Scios 2 DualBeam SEM-FIB (Hillsboro, OR, USA) operated with a 30 keV Ga^+^ ion beam with a final polishing at 5 and 2 keV Ga^+^. TEM lift outs were prepared from the irradiated region.

Characteristics of the He bubbles were calculated using a code built in Python. First, we implemented the filtering stage of mean shift segmentation by taking the output of the function and the filtered “toned” image with color gradients and fine-grain texture flattening. We then converted the color to a gray-scale image, linearly stretching the image to prevent the lack of contrast from representing false objects, and then used a combination of binary and Otsu thresholding to determine the optimal threshold value using Otsu’s algorithm. Afterwards, sequentially, erosion and dilation were applied to the image, which has shown to reduce noise in the output image and increase accuracy of calculations. Next, we used a Euclidean distance transform to calculate the Euclidean distance between the input points in the image. Peaks were then searched for in an image as a coordinate list or Boolean mask, resulting in an arbitrary peak within the same region being returned. The image was then labeled by counting any nonzero features as an input and zero values as background. The watershed algorithm was then employed to separate the overlapping objects in the image. From the preprocessing steps defined above, we can calculate the features of the unique labels defined as He bubbles to draw properties around each region and calibrating the scale accordingly.

## 3. Results

Figure 1 shows the surface composition of the as-fabricated specimens. The NbTa, MoNbTa, and MoNbTaW alloys show no compositional gradients, indicated with the uniform compositional maps of all elements. However, the NBTaVW RHEA shows the microsegregation of V, Nb, and W into intercellular/interdendritic regions, indicated by the regions of increased and decreased intensity in the V, Nb, and W maps. The segregation of V and Nb is consistent with their lower melting temperature compared to Mo, Ta, and W. Appendix A shows the Electron Backscatter Diffraction (EBSD) IPF-Z maps showing the grain structure of fabricated samples.

No changes in the surface composition are observed, nor are any irradiation-induced surface nanofeatures observed on the SEM scale from the He implantation. Appendix A shows SEM-EDS maps of samples following irradiation, showing no change in elemental segregation.

Figure 2 shows the STEM-HAADF micrographs of all samples, with the He bubbles present in the samples varying from 100 to 600 nm beneath the surface, with the He bubbles appearing black. The arrows at the top of the micrographs indicate the location of the sample surface. The images were stitched together from higher-magnification micrographs, and thus stitching artefacts are present in Figure 2. Figure 2f shows the SRIM-predicted He implantation concentration in all samples, peaking around 450 nm beneath the surface, agreeing with experimental results. Clear differences in the size and spatial density of the bubbles are observed between the samples, as evidenced in comparing Figure 2a,b with the much larger bubbles observed in Figure 2b in the NbTa sample. Appendix A shows segmented and thresholded micrographs with bubbles color-coded by depth. A systematic discussion of the bubble size and distribution is presented later in the manuscript. STEM-EDS elemental maps indicate no elemental segregation in the NbTa, MoNbTa, and MoNbTaW alloys, just as in the SEM-EDS. However, the STEM-EDS map for the NbTaVW RHEA sample displayed in Figure 1d shows elemental segregation, which is present on the micron scale before irradiation and is provided in Figure 3. STEM-EDS investigations were performed on all samples to investigate the effects of the composition on the location and size of the He bubbles, these STEM-EDS maps are shown in Appendix A. The W, NbTa, MoNbTa, and MoNbTaW samples show no compositional changes, while only the NbTaVW sample shows elemental segregation in the presence of He bubbles, with V segregation to He bubbles. Figure 3a shows the overlay of the STEM-HAADF and STEM-EDS V elemental map in the He-irradiated region, showing spatial correspondence between the regions of the enhanced V concentrations and the He bubbles. Figure 3b shows the STEM-HAADF micrograph and the STEM-EDS maps from the region beneath the irradiated region where no He bubbles are visible, indicating no nanometer-scale segregation of V.

Figure 4 plots the average He bubble size and the areal density in each sample as a function of the depth beneath the surface. The average size of the He bubbles is largest in the NbTa alloy, followed by the NbTaVW alloy, with the other RHEAs and W having similar average bubble sizes. Additionally, the areal number density of bubbles shows an inverse relationship to size: as the bubbles grow, their density decreases.

When normalized for the sample grain size, the average bubble size and density collapse closer together. The W control sample has the smallest grain size of 275 nm, which may result in the smallest intergranular bubble density. W has a normalized bubble density of 1.44 × 10^−6^ nm^−3^, while the other refractory alloys have higher densities and larger grains. The W control sample has grains elongated parallel to the sample surface, indicative of a rolling process. Slight segregation of bubbles to the interfaces is seen. In the AM alloys, the subgrain structures from solidification and near-surface polishing artifacts are present, although at low angle interfaces, and there is no accumulation of bubbles at interfaces. In the NbTaVW sample, the average grain size is ~24 µm, about 100× larger than the W control sample. When normalized, the bubble density is 5.57 × 10^−8^ nm^−3^. Thus, the grain size dictates the bubble density, increasing the bubble density relative to the grain size through He bubble trapping at the interfaces. The other AM-processed alloys have intermediate grain sizes and bubble densities, indicating that grain size does dictate the bubble size and density, as well as the composition, to be discussed further below.

## 4. Discussion

In nuclear energy systems, there is an intricate interplay between the He bubble size, distribution, and accumulation in materials. Limiting the He bubble size can limit swelling, while limiting the accumulation at the layer boundaries can limit the preferential cracking or delamination sites. An ideal material can limit the He bubble size through the intrinsic microstructure and compositional engineering. In this work, we show that the composition of refractory alloys can dictate the He bubble characteristics.

This work shows that increasing the compositional complexity in the case of RHEAs can increase the radiation tolerance in terms of limiting the He bubble size. Judicious selection of alloying elements can significantly alter the He bubble size and density, although until the sample has four or more elements added, the grain size reduction may play a more important role. It appears that the addition of V to the RHEA composition greatly increases the bubble size, especially when V is replaced by Mo. There are numerous factors that are likely playing a role in dictating the He bubble characteristics, including the melting temperature, the elastic modulus, and the complexity. In all samples, the incident He atoms have enough energy to cause lattice displacements and introduce vacancies (Vac) in the near-surface through the bombardment process. In all refractory elements studied here, the knock-on damage thresholds are well below the He ion energy, and the minimal differences in the amount of displaced atoms as a function of composition are expected. Implanted He tends to bind with vacancies and form He-Vac complexes, which have a high affinity for interstitial He, and which then undergo a trap mutation process to form a bubble [38]. Helium nanobubbles then grow through thermal migration and coalescence of He nanobubbles into larger bubbles. At the temperatures studied here, 900 °C, there is a sufficient supply of vacancies, and combined with the large He implantation nucleation of bubbles, progresses until the growing bubbles become large enough to form strong sinks for the continually implanted He.

In this work, we observe a general decrease in the average size and an increase in the density of the He bubbles as we transition from MoNb, to MoNbTa, to MoNbTaW RHEAs, which is a general increase in the compositional complexity and entropy, mirroring a trend observed by Wang et al. in Ni-based HEAs [39]. The trend of the bubble size in these alloys mirrors the trend in the homologous melting temperature and the elastic modulus. Assuming a Vegard’s law of mixtures of the constituent elements, the melting temperatures increase from 2537 to 2770, to 2948 °C, while the elastic modulus increases from 158 to 197, to 256 GPa [40]. Although, in this work, the high-entropy effect causing a lattice distortion can certainly play a role in altering the dependence of the bubble size strictly on the melting temperature and modulus. As such, in comparison, the pure W sample, which has smaller average bubble diameters (5 nm) than the two and three composition AM-processed refractory alloys, has a melting point of 3417 °C and a modulus of 411 GPa. In the CrMnFeCoNi alloys studied by L. Yang et al., smaller bubbles compared to 304SS and Ni are attributed to a lower point defect mobility in the HEA [41]. The trend towards smaller bubbles as the compositional complexity increases is in contrast to the trend observed by N. Jia et al., who observed large bubbles in a four composition RHEA, with smaller bubbles in three composition RHEA under He irradiation at 700 °C [42]. However, these results become more complex due to the elemental segregation in one of the four-composition alloys, but not in three composition one.

The NbTaVW sample showed larger He bubble diameters than the MoNbTaW sample (13 nm vs. 1 nm), attributed to the substitution of V for Mo. This difference lowers the melting temperature of the alloy to 2860 °C and lowers the elastic modulus to 225 GPa. However, more impactfully, we also observe the accumulation of He bubbles in the NbTaVW RHEA at the V segregated regions. V has the lowest melting point of all elements used to fabricate the HEAs and may segregate out during the fabrication process due to solidification during the part cooling following AM processing. If the V segregation is simply from the fabrication process, then the accumulation of the He bubbles at the V-rich regions may be attributed to the lower vacancy formation energy of V compared to the other elements. T. Seletskaia et al. showed that V also has the lowest binding energy of He to a vacancy of all elements studied here, allowing for rapid He accumulation into larger bubbles [43]. However, Figure 3 shows that the region beneath the irradiation zone did not have the nanoscale elemental segregation seen in the He-irradiated zone. V enrichment was also observed in TiVNbTa RHEAs under 1.5 MeV He bombardment at 700 °C [42]. Elemental segregation during irradiation is commonly due to atomic size differences, as oversized elements preferentially exchange with vacancies [44]. V atoms are the smallest of the four constituent elements; therefore, the segregation of the other elements is not expected. As He is injected into the material, He atoms combine with vacancies, and, in time, nucleate bubbles. As the irradiation fluence progresses, the He bubble nuclei act as sinks for more He atoms and more efficient sinks for vacancies than interstitials [45]. Thus, there is a net flux of vacancies towards bubbles and a net flux of larger atoms (Nb, Ta, W) away from the He bubbles, which results in V enrichment at the bubbles.

In this study, compared to a pure W control sample that was commercially produced, the RHEAs may offer enhanced resistance to the He bubble growth, depending on the composition. However, we must consider the grain size. The pure W sample, as seen in the STEM micrograph, has much finer grains (300 nm vs. 10s of µm for the RHEAs) than the other materials due to the mechanical rolling performed by the manufacturer. Grain boundaries are sinks for He bubbles, and the GBs in the W sample show He bubble accumulation. However, the accumulation of bubbles at the grain boundaries would limit the He inventory within the grains, limiting the size of the He bubbles intragranularly. Thus, the size of the He bubbles in the W sample may be smaller due to the presence of GB. There was no He bubble accumulation at the solidification subgrains observed in the micrographs in the RHEAs. If the RHEAs underwent a similar rolling process to reduce their grain sizes, one could hypothesize that the average bubble size would be smaller due to the presence of an increased grain boundary density. Thus, the limited size and density of the He bubbles in the MoNbTa and MoNbTaW RHEAs may be improved if their grain sizes can be further reduced through a post-fabrication treatment.

While pure W may have the most ideal combination of the He bubble size and density of the materials studied here, there are significant concerns for the usage of W in a fission or fusion reactor, including its ductile–brittle transition temperature, recrystallization, and the ability to fabricate complex components. The performance of additively manufactured RHEAs in other areas relevant for reactors, including fracture properties and recrystallization performance, must be explored further. Future studies will elucidate the mechanical performance and high-temperature performance of the RHEAs studied here, including the potential for self-healing.

Helium irradiation and He bubble formation and accumulation is just one aspect of materials design that should motivate materials choices for future reactors. In this work, we show that compositionally complex (spanning two, three, and four-component alloys) can be developed with AM processing routes. Under exposure to He irradiation, increasing the compositional complexity and entropy may increase tolerance to He irradiation, although ensuring a homogenous microstructure free of elemental segregation may offer the best performance under irradiation. Alloying with V, despite being attractive for use in neutron irradiation environments for its low activation, may be detrimental to HEAs under He irradiation.

## 5. Conclusions

RHEAs are proposed for use in future nuclear fission and fusion reactors, in which they will be exposed to high-temperature irradiation. Their response to He bubble formation should be elucidated to predict their performance in a reactor environment. In this study, we investigate the response of two, three, and four-composition AM RHEAs to 200 keV He irradiation at 900 °C. As the RHEA complexity increases from MoNb, to MoNbTa, to MoNbTaW, the He bubble size decreases, while the density increases. However, when Mo is replaced with V to form a NbTaVW RHEA, V segregation occurs, resulting in the largest He bubbles. The trends of the He bubble characteristics are discussed in relation to the constituent element melting points and elastic moduli, as well as in comparison to a commercial W sample. We show that the performance of additively manufactured RHEAs can be tailored through the precise consideration of the composition to optimize the performance in a helium irradiation environment.

## Figures and Tables

**Figure 1 nanomaterials-12-02014-f001:**
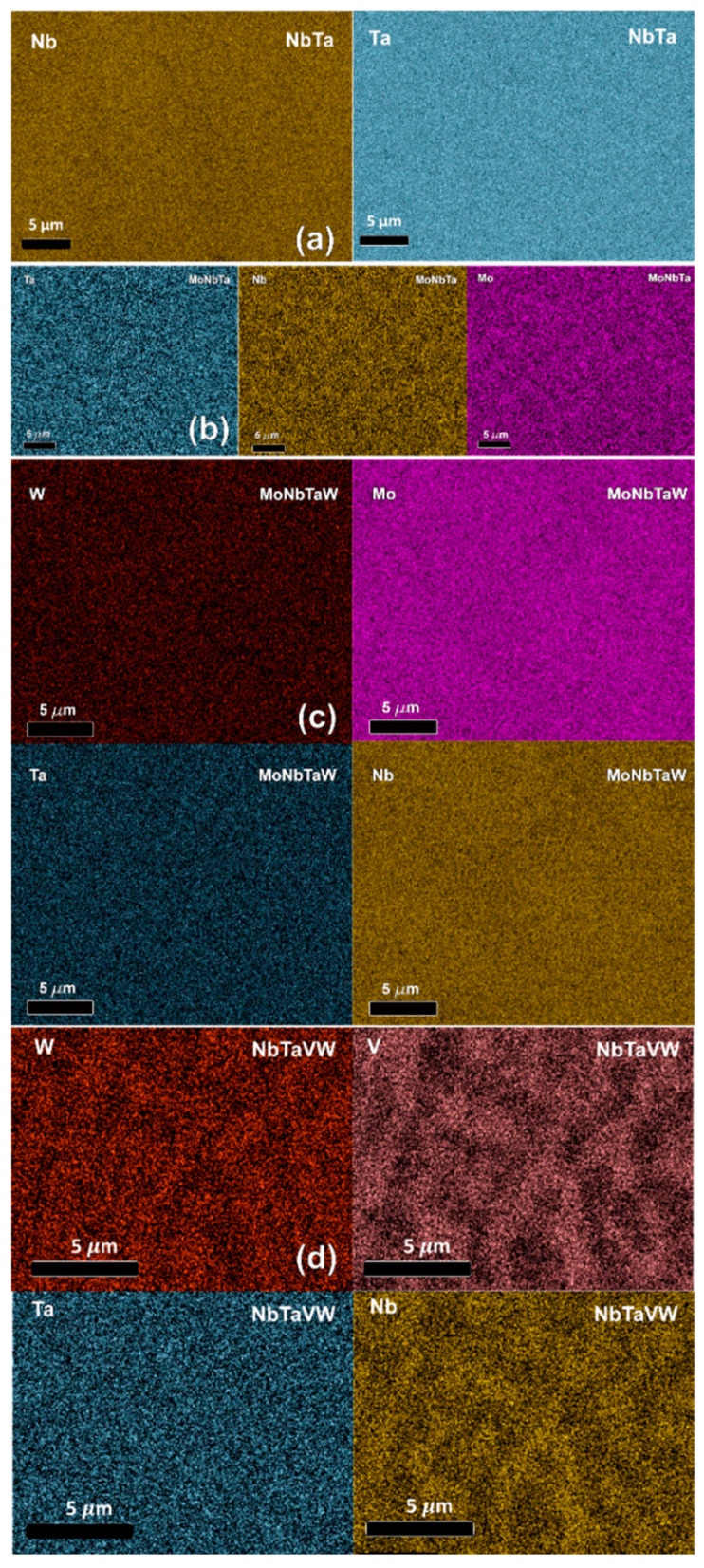
(**a**–**d**) SEM-EDS of AM refractory alloys, as indicated, showing appreciable elemental segregation only in the NbTaVW RHEA.

**Figure 2 nanomaterials-12-02014-f002:**
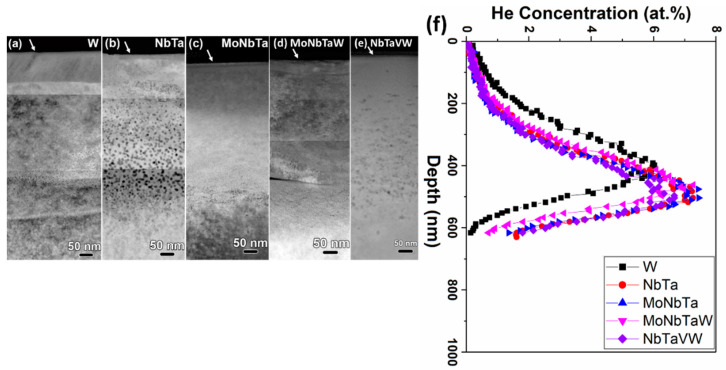
(**a**–**e**) STEM-HAADF micrographs of irradiated samples showing the He bubbles in the subsurface. Arrows indicate the location of the sample surface. (**f**) SRIM-predicted He implantation profiles in all irradiated samples.

**Figure 3 nanomaterials-12-02014-f003:**
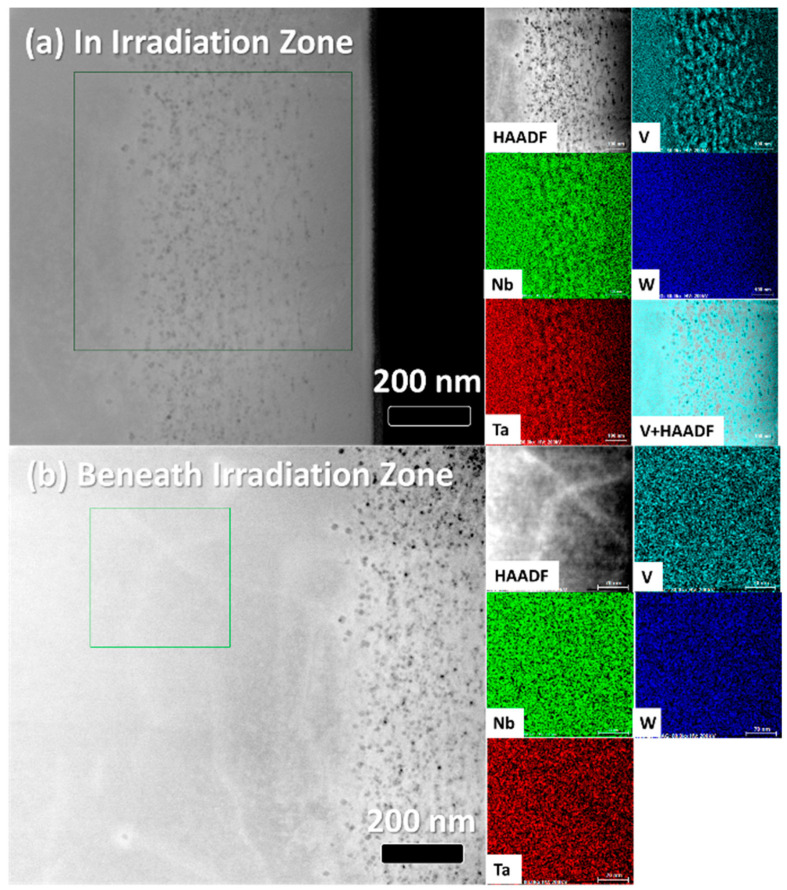
STEM-HAADF micrographs and associated STEM-EDS elemental maps of the NbTaVW sample showing V segregation in the He-irradiated region (**a**), but not in the bulk sample (**b**). Green boxes in the large HAADF micrographs indicate the regions of EDS elemental mapping.

**Figure 4 nanomaterials-12-02014-f004:**
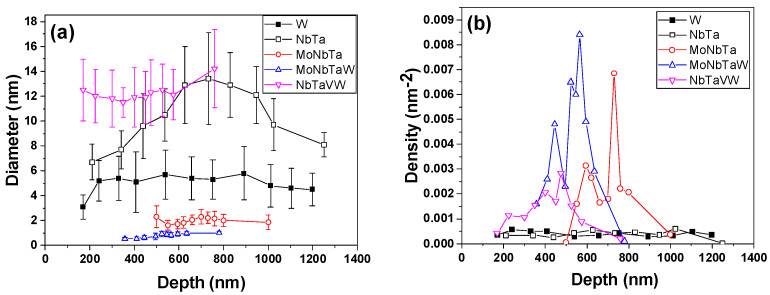
Average He bubble (**a**) diameter and (**b**) density as a function of depth beneath surface for all samples.

## Data Availability

Not applicable.

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
