# Peer review of "Compositional Effects of Additively Manufactured Refractory High-Entropy Alloys under High-Energy Helium Irradiation"

_nanomaterials, 2022, doi:10.3390/nano12122014_

Round 1

Reviewer 1 Report

This work investigates the effect of RHEA composition on the damage (bubble formation) caused by He irradiation.  The manuscript is clearly written and the introduction is well informed.  The results are presented to a high quality and a comparison is made with pure W (not always is a comparison to a conventional alloy made, so it was good to see this).  Observations are appropriately discussed and explained.  The conclusions are supported by the evidence provided.  The work will be of interest to the HEA community.  I have the following further comments: 

  1. There seems to be a repeated section in the Introduction: “Traditional RHEAs are fabricated via sintering or arc/induction melting, which prolongs specimen fabrication time and constricts the design space. Functional materials for future extreme environments require designs tailored for their application, which is not achievable through conventional methods.”
  2. In the interests of transparency (as well as collaboration), the authors might consider making their Python code(s) available to all online (e.g., via GitHub).
  3. Apologies if I missed this, but was the production information for the pure W included? Also, were the bulk compositions of the built parts ever assessed (e.g., using large-area EDS?). 
  4. The maps for the MoNbTa alloy don’t look as uniform as the NbTa results. How do the authors account for this? 
  5. It is stated that “No changes in the surface composition are observed, nor are any irradiation-induced surface nanofeatures observed on the SEM-scale from the He implantation.” Perhaps the results showing this could be included in a supplementary materials document?  Given the samples were held at 900˚C for around an hour whilst being irradiated, segregation or precipitation wouldn’t be surprising.  The same also applies for the STEM-EDS results. 
  6. When Fig. 2 is described in the main text, the fact that the bubbles appear black is stated twice.
  7. Are the STEM-EDS maps shown quantified, or just scaled by number of counts?
  8. The V segregation in Fig. 3 is described at length, but the segregation tendencies of the other elements in the irradiated layer are not mentioned.
  9. How were the sample grain sizes measured?
  10. He-V complexes are mentioned in the discussed – perhaps better to use V only for vanadium, and pick another term for vacancies?
  11. The citation for Jia et al., in the discussion is not provided. Later in the same paragraph, temperature is given with two different units. 

Author Response

Please find responses to reviewer comments in the attachment. 

Reviewer 2 Report

The authors present their work on the compositional effects by comparing the size and density distribution of the helium bubbles in different RHEAs. The most important finding is that increasing the chemical composition of RHEAs can improve the helium bubble resistance of materials according to TEM observations. The results are interesting and important for understanding the helium irradiation damage of RHEAs. It is suitable for publication after some revisions.

The authors use a large number of paragraphs to introduce the generation of helium and the damage of helium irradiation, which is familiar to readers. In the fourth paragraph, the author addressed grain size engineering in the first sentence, yet the cited references mainly illustrate the advantage of an alloying strategy in reducing the helium effect.

Moreover, the best evidence for the main idea of this work is the TEM images and statistics of the size and density of helium bubbles in different RHEAs. However, the images of the helium bubble are only shown in Figs. 2a-e, which is not clear. The black spots, which are defined as bubbles are hard to identifiable in Figs. 2a, c, d and e.

Another question is that the distribution of both the density and the size of the helium bubbles with the depth in Fig. 4 does not correspond to the distribution of the helium concentration in Fig. 2. However, the authors argue that the distribution of helium bubbles in TEM images in Fig. 2 is consistent with the results calculated by SRIM, which is contradictory.

On the third page, the displacement energy of all elements is set as 60 eV during SRIM calculation, is there any reference for this value? If so, please cite the references. As far as I know, the displacement of Tungsten is generally 90 eV. Please refer to “Q. Xu, T. Yoshiie, H.C. Huang, Molecular dynamics simulation of vacancy diffusion in tungsten induced by irradiation, Nucl Instrum. Methods Phys. Res. B 206 (2003) 123-126.”

The first line on page 5: Fig. 2e should be Fig. 1d, please check the similar errors.

Some related references on helium bubbles in metals could be involved in the discussion, such as Acta Mater 226 (2022) 117656, Nano Lett 21 (2021) 5798 and Materials 12 (2019) 1036 etc.

Author Response

(The authors gave the same response as above.)
